# LLM-PBC: Logic Learning Machine-Based Explainable Rules Accurately Stratify the Genetic Risk of Primary Biliary Cholangitis

**DOI:** 10.3390/jpm12101587

**Published:** 2022-09-26

**Authors:** Alessio Gerussi, Damiano Verda, Claudio Cappadona, Laura Cristoferi, Davide Paolo Bernasconi, Sandro Bottaro, Marco Carbone, Marco Muselli, Pietro Invernizzi, Rosanna Asselta

**Affiliations:** 1Division of Gastroenterology, Center for Autoimmune Liver Diseases, Department of Medicine and Surgery, University of Milano-Bicocca, 20900 Monza, Italy; 2European Reference Network on Hepatological Diseases (ERN RARE-LIVER), San Gerardo Hospital, 20900 Monza, Italy; 3Rulex Innovation Labs, Rulex Inc., 16122 Genoa, Italy; 4Department of Biomedical Sciences, Humanitas University, Via Rita Levi Montalcini 4, Pieve Emanuele, 20072 Milan, Italy; 5IRCCS Humanitas Clinical and Research Center, Via Manzoni 56, Rozzano, 20089 Milan, Italy; 6Bicocca Bioinformatics Biostatistics and Bioimaging Centre—B4, School of Medicine and Surgery, University of Milano-Bicocca, 20900 Monza, Italy

**Keywords:** precision medicine, explainable artificial intelligence, primary biliary cholangitis, liver, autoimmunity, risk stratification, machine learning, genomics, genome-wide association study

## Abstract

Background: The application of Machine Learning (ML) to genetic individual-level data represents a foreseeable advancement for the field, which is still in its infancy. Here, we aimed to evaluate the feasibility and accuracy of an ML-based model for disease risk prediction applied to Primary Biliary Cholangitis (PBC). Methods: Genome-wide significant variants identified in subjects of European ancestry in the recently released second international meta-analysis of GWAS in PBC were used as input data. Quality-checked, individual genomic data from two Italian cohorts were used. The ML included the following steps: import of genotype and phenotype data, genetic variant selection, supervised classification of PBC by genotype, generation of “if-then” rules for disease prediction by logic learning machine (LLM), and model validation in a different cohort. Results: The training cohort included 1345 individuals: 444 were PBC cases and 901 were healthy controls. After pre-processing, 41,899 variants entered the analysis. Several configurations of parameters related to feature selection were simulated. The best LLM model reached an Accuracy of 71.7%, a Matthews correlation coefficient of 0.29, a Youden’s value of 0.21, a Sensitivity of 0.28, a Specificity of 0.93, a Positive Predictive Value of 0.66, and a Negative Predictive Value of 0.72. Thirty-eight rules were generated. The rule with the highest covering (19.14) included the following genes: RIN3, KANSL1, TIMMDC1, TNPO3. The validation cohort included 834 individuals: 255 cases and 579 controls. By applying the ruleset derived in the training cohort, the Area under the Curve of the model was 0.73. Conclusions: This study represents the first illustration of an ML model applied to common variants associated with PBC. Our approach is computationally feasible, leverages individual-level data to generate intelligible rules, and can be used for disease prediction in at-risk individuals.

## 1. Introduction

Precision medicine aims to tailor diagnosis, follow-up, and management of individuals based on their genetic and environmental background. Several excellent examples of precision medicine have been produced in the field of cancer and cardiovascular medicine [1,2], while rare liver diseases lag behind.

PBC is a rare disease of the small biliary ducts of the liver [3], characterized by an autoimmune pathogenesis and a strong genetic predisposition, with major histocompatibility complex (MHC) class-II haplotypes and non-MHC loci contributing to the genetic risk [3,4]. Applications of precision medicine in PBC are lacking, likely because it is a complex trait, with dozens of variants participating in the genetic architecture of the disease with a small contribution [3], and also a rare disease, making it more difficult to recruit sufficiently large cohorts for genotyping [5].

Thanks to the establishment of national and international consortia, several genome-wide association studies (GWAS) and two meta-analyses have been published so far, associating 57 variants with PBC [3,6]. The polygenic architecture of the disease, together with the high levels of heritability [3] represent a solid rationale to develop polygenic risk scores (PRSs). PRSs assume that each variant has a linear additive effect on disease [7]; yet, many authors have suggested that PRSs may be limited by their reliance on linear regression [1,8]. 

Machine learning (ML) algorithms can be trained to model the genetic risk of a complex trait, with theoretical advantages related to their ability to handle high-dimensional data by nonlinear effects applied to individual level data [8,9]. ML algorithms can take advantage of prior information—that can be added to a model to train it more effectively—or employ minimal a priori assumptions about the nature of the genetic effects being modelled, potentially also taking into account gene-gene interactions [8,10].

The aim of this study was to evaluate the feasibility and accuracy of an ML-based model for genetic risk prediction of PBC using genome-wide significant variants identified in subjects of European ancestry in the recently published international meta-analysis of GWAS in PBC [6].

## 2. Materials, Methods, and Participants

### 2.1. Study Design and Participants 

All cases met internationally accepted criteria for PBC [11]. The training cohort (hereafter referred as “PRE-RUN”) was composed of 1345 individuals of Italian ancestry: 444 PBC cases and 901 healthy controls; 515 were males and 830 females [6]. The validation cohort (“RUN”) was made up of 834 individuals of Italian ancestry: 255 cases and 579 controls; 335 were males and 499 females [6].

This study included quality-checked, imputed genotype data derived from the recently published international meta-analysis [6]. For a detailed description of the cohorts under analysis, we referred to the methods sections of the meta-analysis by Cordell et al. [6], where the time span of data collection, the collection site and setting, relevant population characteristics, and any inclusion or exclusion criteria used in original studies can be retrieved.

### 2.2. Patient Privacy and Ethical Issues

All participants gave written informed consent for genetic studies. The methods were performed in accordance with relevant guidelines and regulations and approved by San Paolo Hospital. The research conformed to the ethical guidelines of the 1975 Declaration of Helsinki. The protocol was approved by each participating centre in accordance with local regulations [6]. As far as the application of ML to clinical data, the Rulex proprietary software used in this work is compliant with the strictest data privacy regulations, such as the European Union’s General Data Protection Regulation (GDPR). GDPR allows automated ML predictions only if a clear explanation of the logic used to make each decision is provided, which is difficult with black box models. This section of methods adheres to the guidelines and quality criteria for artificial intelligence–based prediction models in healthcare [12]. A full list of the consortium members appears in the Appendix A.

### 2.3. Data Preprocessing

#### 2.3.1. Selection of Genetic Variants

The recent international meta-analysis identified 57 loci [6] associated with PBC at the genome-wide level of significance in patients with European and East Asian ancestry. For the current study, we selected those loci at a genome-wide significant threshold in Europeans (*p* < 5 × 10^−8^) that were reported as more likely to be causal after fine-mapping strategies (Appendix A of the meta-analysis [6]), eventually including 46 non-Human Leukocyte Antigen (HLA) loci (Table 1). 

#### 2.3.2. Import of PLINK Files in the Rulex Environment

Rulex is a novel ML software able to make intelligible predictive models (www.rulex.ai, accessed on 10 September 2022). The Logic Learning Machine (LLM) is the core machine learning algorithm of Rulex (Newton, MA, USA) and represents a method of supervised data mining. Rather than producing a math function, the LLM produces conditional logic rules, fulfilling the definition of explainable Artificial Intelligence (AI) [14], as opposed to deep learning and other “black-box” AI algorithms [15,16]. A list of published works using Rulex in the field of biology and medicine can be found here [17,18,19,20,21,22,23,24]. PLINK files (.map and .ped files) were imported into the Rulex environment by the “Import from text” operator and parsed accordingly. To account for the role of HLA, we selected and extracted from the HLA region only the best associated SNP within each cohort (--assoc function in PLINK). For PRE-RUN, the top variant was chr6:32653792:A:G, while for RUN it was chr6:32429303:A:G. The HLA SNP was recoded as an ordinal variable as follows: homozygous (AA) = 1, heterozygous (AG, GA) = 2, homozygous (GG) = 3. Sex and the pre-selected top HLA variant were used as additional features.

### 2.4. Model Development

#### 2.4.1. Feature Selection

The output of the study was the variable “Status”, identifying whether a subject of the study was a PBC patient (case) or a control. Features with a number of mode values above 95% were removed by the Rulex “Fill/Clean” operator. To avoid redundancy in input features, where redundancy speaks for high correlation, Rulex was used to rank all available features versus the disease status by univariate association based on Cramer’s V. After this univariate ranking, all the features entered a greedy forward selection process. Greedy forward selection is a popular technique for feature subset selection [25]. The main advantages of this approach are its simplicity and its computational scalability, which makes it applicable to many practical problems, including the most complex ones. The algorithm starts with an empty set of features; then, the additional feature is added iteratively to the set, provided that it meets a predefined performance measure.

At each step, the selected feature was the one with the highest correlation with the output, provided that its correlation with any of the already selected inputs did not exceed a threshold value, t. In this way, the landscape of input features was filtered only based on direct correlations (between the input features with the output and among input features with each other). Since our aim was to avoid preliminary multivariate steps before LLM, we avoided the use of indicators such as the Akaike Information Criterion (AIC) [25], since it would implicitly introduce an auxiliary multivariate model by selecting features according to their performance and number.

The second parameter that was used for tuning was the maximum number of features, n, to be selected. The procedure terminated, and no additional features were considered for inclusion if, after a given iteration of the described greedy procedure, n features were included.

#### 2.4.2. Internal Validation

The final list of variants at the end of the feature selection process was used as input for the LLM operator, to build intelligible rules to predict the disease status. The Rulex LLM operator was nested within a process working in cross-validation classifying features associated with the output of interest. For internal validation, LLM operated under a 10-fold cross-validation approach. Cross-validation involves partitioning a sample of data into complementary subsets, performing analyses on one subset (the training cohort) and validating the analyses on the other subset (the test cohort). The time for a complete run (from import to rules generation) was on average ~2.5 h on a working station with 64gb RAM. The output of each run of the model was a list of rules (called ruleset). A metrics section of the pipeline was dedicated to the evaluation of the performance of the rulesets in training and test cohorts.

#### 2.4.3. Hyperparameter Tuning

In machine learning, a hyperparameter is a parameter whose value is used to control the learning process; it can be tuned to change the speed and quality of the learning process.

Several sets of parameters were evaluated before choosing the final optimal model. The parameters undergoing tuning were

-correlation among features: three thresholds, t, based on Cramer’s V value were evaluated (0.7, 0.8 and 0.9);-number of pre-selected and selected features: several fixed combinations of thresholds for pre-selected features and selected features were evaluated (Appendix A);-max error (errmax): errmax represents the maximum level of error for each rule included in the ruleset. In other words, this corresponded to the maximum percentage of cases belonging to output classes different from the predicted one, which verified the rule.

Changes in the performance of the model after modifications of each parameter were evaluated based on the following metrics: Sensitivity, Specificity, Positive Predictive Value, Negative Predictive Value, Accuracy, Matthews coefficient, Youden’s index (definitions of the metrics are reported in the Appendix A).

#### 2.4.4. Final Model Selection and Rules Generation

After 814 runs of the Rulex process, a full list of metrics including all hyperparameters and accuracy metrics was available for choosing the best model. After choosing the combination that maximized the accuracy and specificity of the model, the model was re-run with the chosen set of parameters, and the final ruleset was generated.

The quality of each rule was then evaluated based on the following metrics specific to the Rulex environment: (i) “Covering” is the percentage of samples belonging to the class described by the rule, fulfilling that specific rule; and (ii) “Error” is the percentage of samples belonging to the other classes fulfilling that specific rule. A Feature ranking plot was also generated to help discriminate the most relevant features.

#### 2.4.5. Haplotype Analysis

To investigate whether Rulex LLM identified a risk haplotype, a case-control haplotype analysis was performed in PLINK 1.07 [26]. 

#### 2.4.6. Validation of the Model

To further improve the robustness of our conclusions and reduce the risk of overfitting, we also performed an external validation on a cohort (RUN) that was used only for this purpose. This cohort will be referred to as the forecast or validation cohort.

Since the training and the forecast cohort did not include exactly the same features, only a fraction of the rules of the final model could be applied to the forecast cohort. To minimize the reduction in prediction power of the model due to this heterogeneity, we adopted a multi-step process. We decided not to extract a model in the training cohort based only on the shared features, in order to derive a single model that can be adapted to different forecasting cohorts, enhancing model plasticity and generalization.

We calculated a count of the conditions present in the final ruleset but not verified in the forecast cohort because of missing data. Then, a frequent pattern mining auxiliary layer was built, leveraging the notion that many input features are highly correlated with each other, so that if a variant is missing, another highly correlated variant could ideally be retrieved [25]. More specifically, for each condition, c, the frequent pattern mining branch identified which conditions, among the shared ones, were the most correlated to c. For biological plausibility, only conditions located on the same chromosome as c and not more distant than 500,000 base pairs were considered as candidate conditions for replacement. Let us refer to the condition (among the candidate ones) that is more correlated to c as c’. The all-confidence score was used as a ranking metric for identifying the most correlated condition [27]. Rules constituting the model were adapted by substituting each condition c with c’, provided that the all-confidence score measuring their correlation met a minimum threshold of 0.9.

Finally, the adapted ruleset was applied to the forecasting cohort. The application of this ruleset produced a forecast score, ranging from 0 to 1, for each of the considered individuals. For each subject, the forecast score was initialized to 0.5, and it increased or decreased according to the rules included in the model that the subject meets. For instance, if a patient verified all the rules that predicted to be a case and no rule that predicted to be a control, its forecast score would be 1. Conversely, if the patient verified all the rules that predicted to be a control and no rules that predicted to be a case, its forecast score would be 0. Figure 1 and Figure 2 summarize the pipeline in the training and validation cohorts, respectively.

### 2.5. Statistical Analyses

Comparisons between median scores between cases and controls were performed using the Wilcoxon signed-rank test. Diagnostic accuracy was evaluated using receiving operator characteristic (ROC) curves (R package ROCR). Area under the ROC curve (AUC) is reported together with its 95% confidence interval (CI). Calibration was assessed after calculating risk predictions according to a logistic regression model, which included the continuous forecast score. Individual predicted risks were then divided into ten equally sized categories (i.e., according to deciles). A calibration plot was then produced by comparing the mean predicted risk in each decile (displayed in the x axis) with the observed risk, calculated as the proportion of PBC cases within each decile (displayed in the *y* axis). Brier score, corresponding to the mean squared error of the prediction, was also calculated together with its 95% CI. Calibration analyses were performed with the R package riskRegression. All analyses were performed using R Statistical Software 4.0.3. 

To take into account the linkage disequilibrium (LD) structure of the regions harboring the selected loci (hence avoiding missing genetic information), for each genome-wide significantly associated locus, a region spanning ±250 kb upstream and downstream of the corresponding coding sequence was considered. We hence extracted Single Nucleotide Polymorphism (SNP) genotype data from the imputed set of data using PLINK 1.9 [13].

## 3. Results

### 3.1. Description of the Training Cohort

The training cohort (PRE-RUN) was composed of 1345 individuals of Italian ancestry: 444 PBC cases (37 males and 407 females) and 901 healthy controls (478 males and 423 females). The total number of SNPs in the training cohort was 105,150, distributed along a genomic region of 23,000 kb (46 loci).

### 3.2. Feature Selection

After removing variants with a number of mode values above 95%, 41,899 variants were brought forward into the feature selection process. Univariate association between each of the 41,899 variants and the output (case vs. control) was performed, and features were ranked according to Cramer’s v (Appendix A).

After univariate analysis, an iterative greedy procedure was performed to avoid the inclusion of input features strongly correlated to each other. Different values of hyperparameters (npresel, nsel, errmax, correlation) were evaluated in training and test cohorts (where the test represented the one-tenth randomly selected portion of the training cohort under cross validation). A total of 814 runs of the workflow were performed for hyperparameter optimization (Appendix A). 

The evaluation of the performance of the model was done on the test cohort, to avoid overfitting the model by choosing the best model on the training cohort. The configuration that maximized specificity with a good balance in terms of accuracy and precision in the test cohort had npresel = 872, nsel = 266, errmax = 0.05 and correlation = 0.8. The list of selected variants that entered the classification model as input features is shown in Appendix A.

### 3.3. Final Model

The final LLM model generated 38 rules to classify the disease status. The LLM model reached an Accuracy of 71.7%, a Matthews correlation coefficient of 0.29, a Youden’s value of 0.21, a Sensitivity of 0.28, a Specificity of 0.93, a Positive Predictive Value of 0.66 and a Negative Predictive Value of 0.72 (Appendix A). Covering of rules ranged from 0.45 to 43.28, with a median value of 5.30 (IQR 2.59, 12.61); error ranged from 0.00 to 5.86, with a median value of 3.41 (IQR 1.50, 4.76).

The rule with the highest covering (19.14) predicting PBC was rule 4, with error equal to 3.88. Rule 4 included the following 13 features: female sex AND chr14:92932650 = C AND chr17:43906828 = G AND chr17:43912635 = A AND chr17:44038536 = CA AND chr17:44040823 = C AND chr17:44065263 = T AND chr17:44183317 = C AND chr17:44185431 = T AND chr17:44222335 = G AND chr17:44283022 = A AND chr3:119111870 = T AND chr7:128705730 = T (Table 2). The genes involved in rule 4 were *RIN3*, *KANSL1*, *TIMMDC1*, *TNPO3*. The covering and error of each condition is reported in Appendix A. 

The most informative rule that did not include sex but only genetic variants was rule 11, including seven conditions: chr17:38020058 = AC AND chr17:38049589 = T AND chr17:38070071 = C AND chr17:43933579 = C AND chr2:135188248 = A AND chr2:25332696 = C AND chr3:159726324 = C (Table 3). 

The genes involved in rule 11 were *TNPO3*, *KANSL1*, *TMEM163*, *RARB* and *IL12A-AS1*. The covering and error of each condition is reported in Appendix A.

Feature ranking outlines the most relevant variants that have been used by the LLM to classify the output (Figure 3). 

### 3.4. Haplotype Analysis

To investigate whether Rulex LLM identified a risk haplotype based on the observation that many conditions within the best rule were SNPs located on the same chromosome, a haplotype analysis was performed (Figure 4). Haplotype GACACTCTGA scored as the most frequent (0.515) and was associated with significantly increased risk to develop PBC as compared to GACACCCTGA (0.214, OR 0.8376, 95% CI 0.68–1.03) and AGCTCACAG (0.264, OR 0.6624, 95% CI 0.542–0.809) (*p*-value 0.000205).

### 3.5. Forecast in the Validation Cohort

The validation cohort (RUN) included 834 individuals of Italian ancestry: 255 cases (28 males and 227 females) and 579 controls (307 males and 272 females). In the validation cohort, 74,484 variants were included. 

The number of variants shared between the training and the validation cohorts was 64,918. Since conditions in rules are connected by a boolean AND, if a condition was not found within the validation cohort, the whole rule could not be applied. Out of 139 different unique features presents in rules, 16 (11.5%) were not available in the validation cohort despite the use of the association mining layer. Therefore, 28/38 (74%) rules of the original ruleset were effectively applied in the validation cohort.

The median score in cases was significantly higher than controls (0.52 (IQR 0.50–0.56) vs. 0.43 (IQR 0.28,0.50) (*p* < 0.001)) (Figure 5A); for higher scores, the number of cases increased consistently (Figure 5B). By applying the ruleset derived in the training cohort, the AUC of the model was 0.73 (95% CI 0.69–0.76); the ROC curve in the validation cohort is presented in Figure 5C. In the validation cohort, the LLM model reached an accuracy of 68.5%, a Matthews correlation coefficient of 0.28, a sensitivity of 0.55, a specificity of 0.74, a positive predictive value of 0.48 and a negative predictive value of 0.79. The LLM model produced individual predicted risks that were in strong agreement with the observed risks, indicating a good level of calibration (Appendix A).

## 4. Discussion

Our proof-of-concept study shows that an ML-based model using common genetic variants to predict genetic susceptibility to PBC is (1) computationally feasible; (2) methodologically innovative; (3) accurate and well calibrated. More specifically, the Rulex workflow generates the final ruleset within three hours, provides intelligible if-then rules for prediction, and achieves good accuracy in discriminating between PBC cases and controls in a new validation cohort. 

From a result assessment perspective, it would have been possible to introduce a wrapper-based feature selection technique, possibly driven by the Akaike criterion. Yet, we set up a simpler, filter feature selection process (such as the ones discussed, for instance, in [28]), so that the Logic Learning Machine model could be evaluated also in terms of its implicit multi-variate feature selection capabilities. 

Our findings are important because they translate the most updated knowledge about the predisposing genetic variants associated in previous GWAS to PBC in applicable rules that can be used for risk stratification.

In terms of clinical translation, the low prevalence of PBC in the general population makes mass screening neither feasible nor cost-effective: the target population for SNP genotyping and applications of rules for prediction would be at-risk individuals, such as first-degree relatives of PBC patients, who are known to have increased risk of developing the disease [29]. A recently presented congress communication [30] reported that, in a prospective cohort of first-degree relatives of patients with PBC, the prevalence of PBC was 5 cases out of 231 individuals assessed (2%). The prevalence of PBC-specific autoantibodies in first-degree relatives was 28 out of 231 (12%), with a higher prevalence in sisters (23%). If these data were incorporated into our analysis, the final model would have a likelihood ratio of 4.0, meaning that if a first-degree relative tested positive, the risk of PBC would increase from 2% to 27% (2% + 25%) [31]. Clinical strategies to deal with first-degree relatives are still in their infancy. Our LLM model could represent a valuable tool to allocate attention and resources across individuals with higher levels of genetic risk [32].

Regarding the methodology employed, there are some innovative aspects that are worth mentioning. First of all, the pipeline presented in this study utilized individual data and not summary statistics. Rulex did approach genetic information in a different manner than standard statistical genetic methods do: it did not evaluate a statistical imbalance of the allele frequency of a variant between cases and controls, but rather how the concomitant presence of genetic variants having a specific DNA base would associate to discriminate between cases and controls. This approach is innovative and makes our model different from a PRS. For instance, the rule with the highest covering included only one SNP among those most significant based on effect sizes and p-values derived from previous studies; in other words, univariate statistical significance was not the only parameter to consider for the ML model to predict disease status.

Multivariate computational approaches such as ours may be able to capture the complex relationships among risk variants for complex traits, which represents a possible innovation for the field [8]. Groups instead of single attributes were linked to case/control prediction by Rulex LLM, indirectly inferring also complementary relations among inputs. Although our analysis was not aimed at studying epistasis specifically, the boolean AND that links conditions within the same rule might represent a proxy for statistical epistatic interaction. Rulex LLM could represent a novel method to improve the way gene–gene interactions are taken into account [33]. Further studies are needed to assess the applicability of Rulex LLM to study epistatic interactions.

In terms of model explainability, the LLM generates if-then rules that are easy to understand. There is growing awareness in the scientific community about the importance of model explainability when ML algorithms are applied in the biomedical field [14,15]. There is also increasing concern related to the possible risk of gender and racial discrimination enhanced by ML algorithms, and this could be more problematic when the user cannot recognize how the algorithm is generating the output [34]. The explainability of the LLM rules makes the algorithm particularly interesting for future applications in risk stratification, as compared with black-box models like deep learning algorithms characterized by excellent accuracy counterbalanced by low levels of explainability [15,35]. The high expectations behind ML should not preclude the recognition that ML algorithms have both pros and cons, and their use should be applied following international data protection and ethical guidelines for AI applications [36].

Our work has some limitations that need to be acknowledged. We did not perform an analysis comparable to a GWAS, because we pre-selected top variants from the international meta-analysis [6], both for computational reasons (a smaller collection of variants was deemed more suitable for a proof-of-concept study) and to follow what has been historically performed for PRSs (pre-selection based on p-value). This approach could have been over-conservative; the pre-selection of top variants may have affection the capability of Rulex to leverage variants with lower effect size to make the model more robust and with higher covering. The next step will be to expand the analysis to the whole genomes (including sexual chromosomes [37]) to assess the scalability of our pipeline. We might anticipate that LLM rules would employ as conditions some variants that would not reach the established genome-wide threshold of significance by standard methods, since they might be important for classification based on the non-linear association method behind LLM. In addition, the greedy feature selection process may have been quicker than other strategies such as exhaustive grid search processes, at the expense of robustness. Iterative strategies such as gradient evolution algorithms should also be tested to understand whether accuracy can be improved while keeping complexity, and consequently the computational time, controlled [38]. 

Advocates of PRSs do affirm that they will be more diffuse than ML-models based on individual data [7]; the main reason behind this statement is that PRSs use summary statistics and do not need individual data, overcoming ethical and logistic limitations related to genetic data sharing. We acknowledge this limitation, underlining that PRS and our ML approach are complementary.

## 5. Conclusions

To conclude, our study represents the first illustration of a successful analysis of common genetic variants with ML to study the genetic liability of a rare liver disease such as PBC. ML is computationally feasible and generates accurate information that can be leveraged for disease prediction in at risk individuals. Our work paves the way for future prospective studies targeting relatives of patients with PBC and aiming at more intensive follow-up for early identification and timely treatment of new PBC cases.

## Figures and Tables

**Figure 1 jpm-12-01587-f001:**
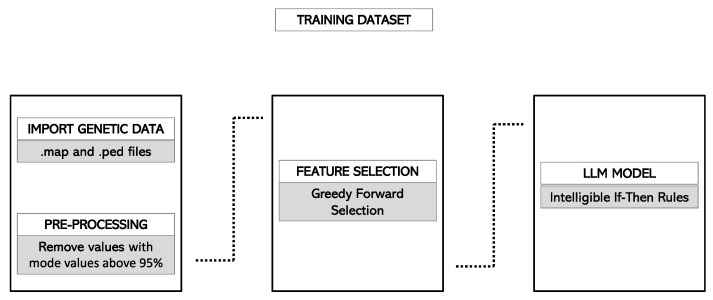
Classification pipeline in the Rulex environment. Information is imported and parsed and then entered into a feature selection branch working in cross-validation. Several hyperparameters are evaluated, and their value can be tuned. Finally, the LLM model generates if-then rules for classification. After completing the pre-defined number of runs, the process generates a number of metrics, including accuracy measures and the feature ranking of attributes. Abbreviations: LLM, Logic Learning Machine.

**Figure 2 jpm-12-01587-f002:**
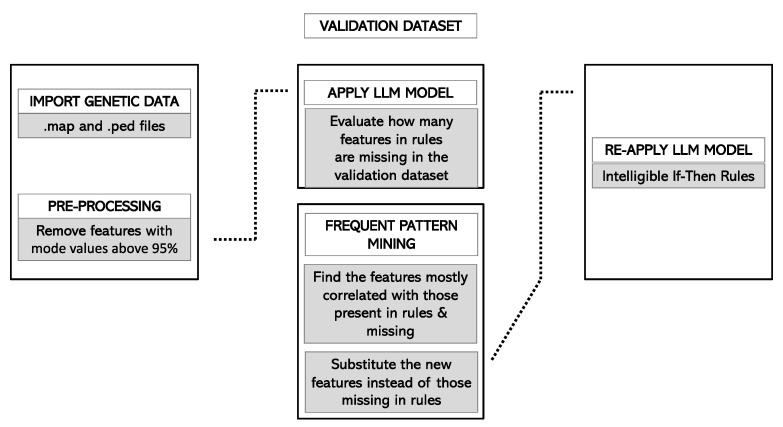
Forecast pipeline in the Rulex environment. Information is imported for the validation cohort. The LLM model is applied to the validation cohort. The conditions that are missing in the validation cohort are analyzed, and a dedicated association mining algorithm operates to find in the training cohort new features that are present in the validation cohort and are correlated (at a predefined threshold) to the missing conditions. The LLM model is then re-run in the validation cohort, and metrics of accuracy are calculated. Abbreviations: LLM, Logic Learning Machine.

**Figure 3 jpm-12-01587-f003:**
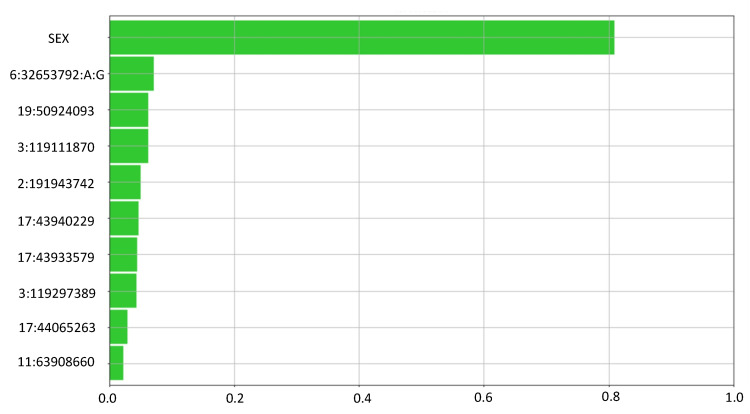
Feature Ranking summary. The Feature Ranking task computes a set of measures to assess the relevance/usefulness of the input attributes within the LLM rules. Absolute relevancy gives an aggregate measure of how “strong” the correlation is between a given input attribute and the output. As expected by the known female predominance of the disease, sex ranked first, and its relevance was 0.81, 11.5 times more relevant than the second feature (the HLA SNP) and 13.5 times more relevant than the third one (the first non-HLA SNP). Among genetic variants, the HLA SNP ranked second, with a relevance of 0.07. Among non-HLA variants the best ones were chr19:50924093 (SPIB) (0.06), chr3:119111870 (TIMMDC1) (0.06) and chr2:191943742 (STAT4) (0.05).

**Figure 4 jpm-12-01587-f004:**
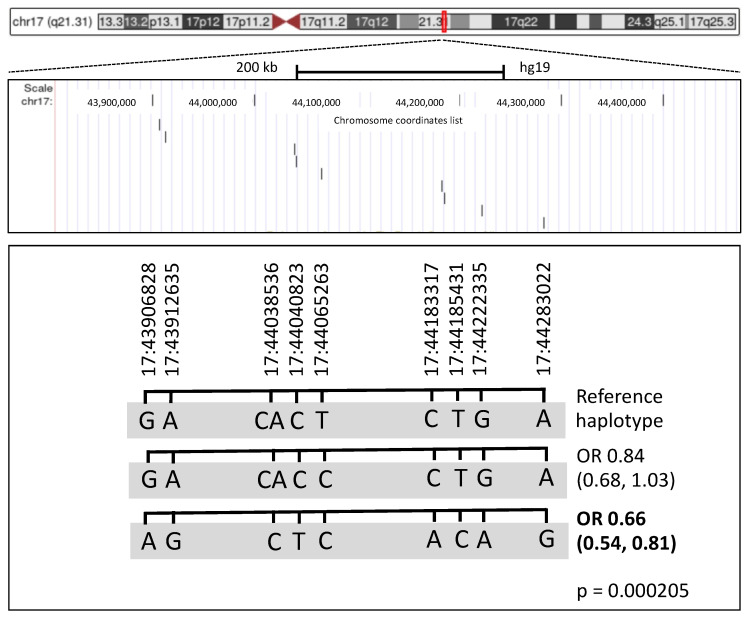
Haplotype reconstruction based on SNPs identified as main conditions by Rulex. The genomic region with position of the 9 SNPs/conditions identified by Rulex. On the top, the ideogram of the entire chromosome 17 is shown; the close-up view indicates the relative positions of the SNPs projected in the lower part of the figure, where haplotypes are listed. Haplotypes were reconstructed using the Plink software; odds ratios (OR) and 95% confidence interval (in parenthesis) were calculated relative to the reference haplotype.

**Figure 5 jpm-12-01587-f005:**
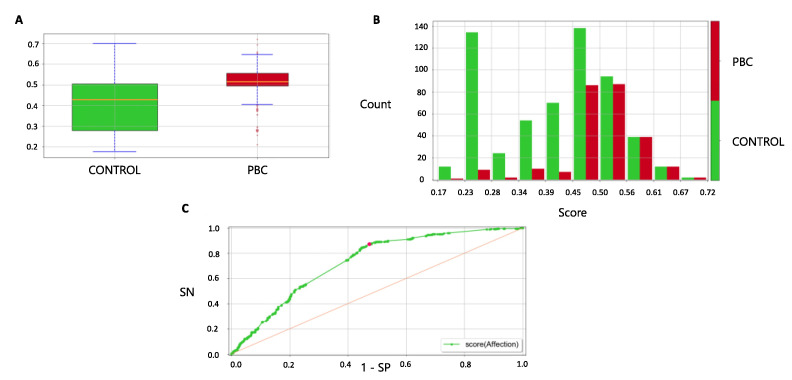
Model performance in the validation cohort. (**A**) Score distribution between cases and controls in the validation cohort. In the box plot, boxes define the interquartile range; thick lines refer to the median. The *p*-value was calculated using the Wilcoxon rank sum test. Legend: 1 = healthy control (green color), 2 = PBC case (red color). (**B**) Proportion of cases and controls by deciles of score in the validation cohort. Legend: 1 = healthy control (green color), 2 = case (red color). (**C**) ROC curve for case-control discrimination in the validation cohort. AUC, Area Under the Curve; PBC, Primary Biliary Cholangitis; ROC, Receiver Operating Characteristics.

**Table 1 jpm-12-01587-t001:** Selected loci at genome-wide significant threshold in Europeans.

Chr	Gene	Start	End
1	*MMEL1*	2,273,723	2,773,723
1	*IL12RB2*	67,570,194	68,070,194
1	*CD58*	116,815,083	117,315,083
1	*FCRL3*	157,420,290	157,920,290
1	*DENND1B*	197,530,966	198,030,966
1	*CACNA1S*	200,769,059	201,269,059
2	*DNMT3A*	25,264,333	25,764,333
2	*TMEM163*	135,091,200	135,591,200
2	*STAT4*	191,693,742	192,193,742
3	*PLCL2*	16,711,265	17,211,265
3	*RARB*	25,133,587	25,633,587
3	*TIMMDC1*	118,969,934	119,469,934
3	*IL12A-AS1*	159,410,283	159,910,283
4	*NFKB1*	103,290,780	103,790,780
4	*TET2*	105,878,954	106,378,954
5	*IL7R*	35,631,130	36,131,130
5	*LOC285626*	158,509,900	159,009,900
6	*OLIG3*	137,723,068	138,223,068
7	*ITGB8*	20,128,801	20,628,801
7	*ELMO1*	37,132,465	37,632,465
7	*TNPO3*	128,367,466	128,867,466
7	*ZC3HAV1*	138,479,543	138,979,543
9	*HEMGN*	100,491,912	100,991,912
10	*WDFY4*	49,775,396	50,275,396
11	*DEAF1*	396,986	896,986
11	*CCDC88B,*	63,860,422	64,360,422
11	*POU2AF1*	110,989,365	111,489,365
11	*DDX6*	118,490,104	118,990,104
12	*TNFRSF1A*	6,190,009	6,690,009
12	*ATXN2*	111,657,431	112,157,431
13	*LINC02341*	42,805,002	43,305,002
13	*DLEU1*	50,561,220	51,061,220
14	*RAD51B*	68,499,927	68,999,927
14	*RIN3*	92,864,787	93,364,787
14	*EXOC3L4*	103,314,807	103,814,807
16	*CLEC16A*	10,924,365	11,424,365
16	*IL4R*	27,153,469	27,653,469
16	*DPEP2*	67,786,939	68,286,939
16	*LOC105371388*	85,769,271	86,269,271
17	*ZPBP2*	37,794,893	38,294,893
17	*KANSL1*	43,899,348	44,399,348
18	*CD226*	67,276,026	67,776,026
19	*TYK2*	10,225,652	10,725,652
19	*MAST3*	17,985,882	18,485,882
19	*SPIB*	50,676,742	51,176,742
22	*RPL3*	39,490,078	39,990,078
Total	46	23,000 kb	

Selected loci at the genome-wide significant threshold in Europeans. Second and third columns report coordinates of the analyzed genomic region (hg38, start and end, respectively). Abbreviations: Chr, Chromosome. To take into account the linkage disequilibrium (LD) structure of the regions harboring the selected loci (hence avoiding missing genetic information), for each genome-wide significantly associated locus, a region spanning ±250 kb upstream and downstream of the corresponding coding sequence was considered. We hence extracted Single Nucleotide Polymorphism (SNP) genotype data from the imputed set of data using PLINK 1.9 [13].

**Table 2 jpm-12-01587-t002:** Best rule for prediction including sex as condition.

Id Rule	4
Number of conditions	13
Output attribute	Affection
Output value	Case
Covering %	19.14
Error %	3.88
Condition 1	14:92932650_2 = “C”
Condition 2	17:43906828_1 = “G”
Condition 3	17:43912635_1 = “A”
Condition 4	17:44038536_1 = “CA”
Condition 5	17:44040823_1 = “C”
Condition 6	17:44065263_1 = “T”
Condition 7	17:44183317_1 = “C”
Condition 8	17:44185431_1 = “T”
Condition 9	17:44222335_1 = “G”
Condition 10	17:44283022_1 = “A”
Condition 11	3:119111870_1 = “T”
Condition 12	7:128705730_1 = “T”
Condition 13	Sex = F

“Covering” is the percentage of samples belonging to the class described by the rule fulfilling that specific rule; “Error” is the percentage of samples belonging to the other classes fulfilling that specific rule.

**Table 3 jpm-12-01587-t003:** Best rule for prediction using only genetic information.

Id Rule	11
Number of conditions	7
Output attribute	Affection
Output value	Case
Covering %	12,162162
Error %	4,661487
Condition 1	17:38020058_2 = “AC”
Condition 2	17:38049589_2 = “T”
Condition 3	17:38070071_2 = “C”
Condition 4	17:43933579_1 = “C”
Condition 5	2:135188248_1 = “A”
Condition 6	2:25332696_2 = “C”
Condition 7	3:159726324_1 = “A”

“Covering” is the percentage of samples belonging to the class described by the rule, fulfilling that specific rule; “Error” is the percentage of samples belonging to the other classes fulfilling that specific rule.

## Data Availability

Data can be shared only after a research proposal for their use is provided to the Italian PBC Genetic consortium (email to pietro.invernizzi@unimib.it), approved and a data transfer agreement is signed according to GDPR rules. The pipelines presented in this work can be made available upon reasonable scientific interest and should be operated in the Rulex environment; to download the software and obtain a license, an application for academic purposes should be sent to Rulex Data Science group (email to damiano.verda@rulex.ai).

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
