# Peer review of "LLM-PBC: Logic Learning Machine-Based Explainable Rules Accurately Stratify the Genetic Risk of Primary Biliary Cholangitis"

_jpm, 2022, doi:10.3390/jpm12101587_

Round 1

Reviewer 1 Report

A very well written and interesting study.  The applicability of the strategy remains unclear but the idea is a good one and the methods sound.  

Author Response

We are grateful for the positive comments of the reviewer. We do acknowledge that our study is still preliminary in nature and a proof-of-concept, but we envisage a possible application of the most significant rules by creating a panel of variants included in the rule and sequencing first-degree relatives of patients with PBC to provide a risk assessment.

Reviewer 2 Report

I enjoyed reading this paper, on an emerging field and using an innovating methodology, providing new promising data about the  genetic evaluation of the risk of PBC that may encourage new further approaches for clinical follow up, better pathogenic understanding,  or even new treatment strategies. I found it relatively easy to read despite the complexity of the study and some difficulties I had with some words, as I describe later.   The Introduction, Material and Methods, Results, Discussion and Conclusions are equilibrated and correspond appropriately to the aim.     Line 33 Feature: I suggest to describe a more accurate description like “genetic variants” or any other of your preference, before using “feature” to the end of the writing.   Line 34: About "phetotype”:if PBC (presence or absence) is the only phenotype characteristic, substituting phetotype by PBC may be more accurate and better understandable.   Line 36: About “set”: A recomendation to use the same word (in that case cohort, related only to people) throughout the article for the groups of patients and/or healthy controls, and limit the use of “set” to groupimgs that does not include people.   Line 79: “international meta-analysis of GWAS in PBC”: refer to reference 6 alone or to both 3 and 6? The addition of the citation numbers may clarify it.   Line 80: Addition of “Participants” to the title “Materials and methods”, if posible.   Line 82: criteria for PBC [11]: reference [11] is the 2009 PBC criteria paper, but  a new edition was published in 2019. Could you add a comment about this?   Line 110: About the variant selection detailed at “2.3.1 Selection of genetic variants”, 14 out of the 22 variants listed on table 2 of  the Cordell et al study are below the threshold (p<5*10-8). However, only 11 are listed at table 1 of your study and 3 of the 14 (INAVA/1q32.1, TRIM14/9q22.33, IRF7/11q15.5) are not. Therefore “almost” must be added between “selected” and “those”, and also the reason why some were not included (if it is not explained anywhere else).   Line 172: A short definition of hyperparameters will facilitate article understanding to non experts readers.   Line 264: the data described here are on table 1 that may be re-cited here, at the end of the paragraph, if possible.   Line 302: "did not include only genetic information", was not celar enough. If this means "the only rule that does not include female sex condition is rule 11", naming directly "female sex" condition will be easier for readers.  

Line 313:  Figure 3: the size of the characters that describe de X and Y axis must be increased for better Reading.

Line 329: Figure 4: the size of the characters that describe de haplotype content (top rectangle) and the chromosome coordinales list (second rectangle)  must be increased for better Reading.

Line 356: Figure 5: the size of the characters and the magnitudes adscribed to the X and Y axis must be increased for better reading

Author Response

We are grateful for the positive comments of the reviewer. 

We have amended all the wording suggestions. The old wording is strikethrough, the new one is marked in red color. We increased the magnitude of the characters in figures as much as possible to improve the readability. 

As regards the suggestions that required a comment:

-We reported the old version of the PBC diagnostic criteria because the recruitment occurred around 2009. Yet, the diagnostic criteria has not changed so that one should not infer that a new recruitment based on new diagnostic criteria would bring different clinical phenotypes under analysis.

-On selection of genetic variants: we have actually selected those genes reported in the Suppl. Table 3 of Cordell et al 2021 because we thought that they were better candidates having gone through the process of fine-mapping. We have added this point in the methods section and thank the reviewer for allowing us to clarify better our feature selection process.

Attached the revised version of the manuscript.

Reviewer 3 Report

I congratulate the authors about this highly relevant work. I have a few suggestions that limit to two. The first is the terminology of OLD-IT and New IT should be changed as it has been discussed in some AI meetings. It may be important not to label with old and new, but with pre-RUN and RUN or something similar. The Akaike criterion should have been implemented. Perhaps, the authors should discuss this aspect in the relevant session in the discussion.

Author Response

We are grateful for the positive comments of the reviewer. 

We have amended the wording suggestions.

From a result assessment perspective, it would have been possible to introduce a wrapper-based feature selection technique, possibly driven by the Akaike criterion. Yet, we set up a simpler, filter feature selection process (such as the ones discussed, for instance, in [Bommert, A.; Sun, X.; Bischl, B.; Rahnenführer, J.; Lang, M. Benchmark for filter methods for feature selection in high-dimensional classification data. Comput. Stat. Data Anal. 2020, 143, 106839, doi:https://doi.org/10.1016/j.csda.2019.106839]), so that the Logic Learning Machine model can be evaluated also in terms of its implicit multi-variate feature selection capabilities. We have added this point to the discussion.
